# The Emergence of New Aggressive Leaf Rust Races with the Potential to Supplant the Resistance of Wheat Cultivars

**DOI:** 10.3390/biology10090925

**Published:** 2021-09-16

**Authors:** Reda Ibrahim Omara, Yasser Nehela, Ola Ibrahim Mabrouk, Mohsen Mohamed Elsharkawy

**Affiliations:** 1Wheat Diseases Research Department, Plant Pathology Research Institute, Agricultural Research Center, Giza 12619, Egypt; amr.khader@agr.kfs.edu.eg (R.I.O.); ola_pathology@yahoo.com (O.I.M.); 2Department of Agricultural Botany, Faculty of Agriculture, Tanta University, Tanta 31511, Egypt; yasser.nehela@ufl.edu; 3Department of Plant Pathology, Citrus Research and Education Center, University of Florida, 700 Experiment Station Rd., Lake Alfred, FL 33850, USA; 4Agricultural Botany Department, Faculty of Agriculture, Kafrelsheikh University, Kafr Elsheikh 33516, Egypt

**Keywords:** wheat, *Puccinia* *triticina*, physiological races, cluster analysis, *Lr* genes, molecular markers

## Abstract

**Simple Summary:**

The pathogen that causes wheat leaf rust, *Puccinia triticina*, possesses numerous aggressive races that can erode the resistant genes in its host. This study presents the recognition of the new physiological races of *P. triticina*, their distribution, and their resistance genes in wheat cultivars, which are critical for directing and improving wheat breeding programs for resistance to leaf rust. Winds often transport the pathogen’s initial inoculum from one country to another. Our findings trigger an alert to the whole world about developing races capable of supplanting leaf rust resistance.

**Abstract:**

Characterization of the genetic structure and the physiological races of *Puccinia triticina* is a growing necessity to apply host genetic resistance against wheat leaf rust as a successful control strategy. Herein, we collected and identified about 130 isolates of *P. triticina* from 16 Egyptian commercial wheat cultivars grown at different locations, over two seasons (2019/2020 and 2020/2021). The 130 isolates of *P. triticina* were segregated into 17 different physiological races. TTTST and TTTKS were the most common virulent races, whereas TTTST and MTTGT were the most frequent races. The races were classified into three groups, based on their distinct DNA band sizes (150 bp, 200 bp, and 300 bp) after RAPD analysis. The new wheat cultivars (Sakha-94, Sakha-95, and Shandweel-1) infected with the most virulent race (TTTST), Gemmeiza-12, and Misr-3 were resistant to all physiological races. The resistance of these cultivars was mostly due to the presence of *Lr19-* and *Lr28-*resistant genes. Our results serve as a warning about emerging aggressive races capable of supplanting resistance to leaf rust, and help in the understanding of the pathotype–cultivar–location association and its role in the susceptibility/resistance of new wheat cultivars to *P. triticina.*

## 1. Introduction

Bread wheat (*Triticum aestivum* L.) is a globally important cereal crop. It is widely produced, and leads the grain market along with corn and rice [1]. Wheat production in Egypt has increased during the last decade, from 7.2 million metric tons (MMT) in 2010 to 8.9 MMT in 2020 (approximately 23.6% comprehensive increase) [2,3]. Briefly, the most recent forecast of the USDA-Foreign Agricultural Service (FAS-Cairo) estimated that Egypt’s wheat production would reach 9.0 MMT during the 2021 marketing year, which reflects about a 1.12% increase compared to 8.9 MMT in 2020 [4]. However, they also forecast an approximately 1.53% increase in Egypt’s wheat imports (from 13.0 MMT in 2020 to 13.2 MMT in 2021) [4].

Egyptian wheat production has improved through increasing yields by unit area via multidisciplinary breeding, cultivation, and pest management techniques that include, but are not limited to, utilization of high-yielding varieties, intensifying the distribution of the certified seeds to growers, implementation of ideal agricultural practices (sowing time, laser leveling, fertilization, etc.), and increasing the cultivated area [4]. However, wheat production struggles with several biotic and abiotic challenges that usually cause major economic losses in terms of yield and quality. For instance, pests and diseases were estimated to cause about 21.5% yield loss at a global level and per hotspot for wheat (ranged from 10.1–28.1%) [5]. Among the about 31 pests and phytopathogens described in wheat, fungal diseases, particularly rust diseases, cause the most serious losses [5,6].

Wheat leaf rust, caused by the basidiomycetous fungus *Puccinia triticina* Erikss (Pucciniomycetes; Pucciniales; Pucciniaceae), is one of the most prevalent and extensively spread persistent diseases, and causes considerable annual losses in grain yield in most wheat-producing countries [7,8,9]. In Egypt, grain yield losses due to leaf rust infection in some susceptible wheat cultivars reached 23%, depending on the severity of the pathogen and the stage of crop growth at the time of infection [10,11]. Moreover, numerous new wheat cultivars were removed and discarded soon after their introduction and widespread usage in agriculture due to their susceptibility to leaf rust disease [12].

Subspecies, which is a genetically distinct population, is often referred to as race. *P. triticina* is a heteroecious phytopathogen that requires two distinct hosts. Briefly, it spends its Telial/uredinial stages on the primary host(s), including *T. aestivum*, *T. turgidum* var. durum, *T. dicoccon*, and *T. dicoccoides*; however, it spends its pycnial/aecial stages on alternate host(s), which mainly includ *Thalictrum speciosissimum* (synonym *T. flavum glaucum,* Yellow Meadow-rue) and *Isopyrum fumaroides* [13]. Fortunately, alternate hosts of *P. triticina* do not existed in Egypt and its urediniospores cannot persist or survive in Egypt during the summer; nevertheless, the source of a primary inoculum is typically brought in by northern winds from foreign sources annually [12,14]. Collectively, all these facts suggest that leaf rust populations in Egypt are made up of a wide range of races [15,16].

Wheat leaf rust epidemics in Egypt are sporadic. This might be mainly due to the variations in weather conditions, where the dry climate in some governorates makes the disease less virulent than in other areas. However, wheat-cultivated governorates with favorable environmental conditions are more suitable for the spread of the disease and might be considered hotspots of leaf rust disease [11,17,18]. Furthermore, in the presence of favorable climatic conditions, many areas of the country may see a fast rise and spread of this disease, with virulent isolates being detected throughout a large geographic area [19].

Growing resistant cultivars is the most efficient, economically profitable, and eco-friendly strategy to control *P. triticina*; however, wheat breeding for resistance to leaf rust disease is complicated because of the ability of *P. triticina* to defeat host resistance. Although about 75 genes have been identified in wheat cultivars resistant to *P. triticina*, the majority are ineffective against current *P. triticina* races [20]. Due to sexual recombination and mutations, new pathotypes of *P. triticina* are continuously emerging that provide a massive array of virulence to this pathogen and allow it to withstand newly deployed resistance genes in commercial cultivars. In seedling and adult plant resistance to leaf rust in wheat, more than 75 leaf rust resistance (*Lr*) genes have been identified, recognized, and reported [21,22]. Most *Lr* genes come from hexaploid bread wheat or wheat-related wild grass species, with just a few being discovered and described in tetraploid durum wheat [23]. Several race non-specific *Lr* genes, including *Lr*34 and *Lr*67, have been discovered, mostly in adults, providing resistance to various pathogen species [24]. Building a strong and updated knowledge base about the effectiveness of *Lr*-genes and virulence dynamics of *P. triticina* populations is critical for successful wheat breeding for resistance to leaf rust. Moreover, consistent monitoring and timely detection of new phenotypes of *P. triticina* play a key role in the adjustment of these breeding programs.

Annual virulence surveys are essential for understanding the evolution and emergence of new virulent races, as well as virulence changes in race physiology within a population [25]. Therefore, identifying, characterizing, and developing new sources of resistance is a growing necessity [26].

According to the study done by Kolmer and Liu [27], virulence infection types of isolates to wheat lines with resistance genes varied substantially for a random amplified polymorphic DNA (RAPD) distance, indicating a general connection between virulence and RAPD phenotype. The similarity of *P. triticina* isolates’ genetic backgrounds, as determined using RAPD markers, suggests that the observed variations in pathogenicity may have resulted from selection for particular virulence matching to resistance genes in wheat varieties cultivated in the area. The amount of variation found among isolates varied across pathogen groups, but the main differences were constant [28].

In the current study, our objectives were to define the physiological races of leaf rust that overcome resistance in Egyptian commercial wheat cultivars and to understand the pathotype–cultivar–location association and its role in susceptibility/resistance of new wheat cultivars to *P. triticina*. Herein, we studied the geographic distribution of *P. triticina* populations on 16 Egyptian commercial wheat cultivars, in four main wheat-growing governorates in Egypt, during two successive seasons (2019/2020 and 2020/2021). The identification of current, as well as new, physiological races of *P. triticina* and their resistance genes in wheat cultivars is critical for directing and improving wheat breeding programs for resistance to leaf rust, as well as understanding field reactions of new cultivars to the most current *P. triticina* populations.

## 2. Materials and Methods

The experiments were performed in four different governorates, Kafrelsheikh, Beheira, Sharkia, and Alexandria, as well as in the leaf rust greenhouse at the Wheat Diseases Research Department, Plant Pathology Research Institute (PPRI), Agricultural Research Center (ARC), Giza, Egypt, during successive 2019/2020 and 2020/2021 seasons. In addition, molecular analysis was performed at the Molecular Biology Laboratory, Faculty of Agriculture, Cairo University.

### 2.1. Race Analysis

#### 2.1.1. Survey

An annual survey of leaf rust was performed to collect infected leaves showing typical symptoms. Wheat fields and the Egyptian Wheat Trap Rust Nursery (EWTRN) in the governorates of Kafrelsheikh, Beheira, Sharkia, and Alexandria were utilized for sampling. The Trap Nursery included 130 wheat entries which were planted in 2-m long rows and were 30 cm apart, with a seed rate of 5 g per entry. These entries included 16 wheat cultivars, monogenic lines for stripe, leaf, stem rusts, and highly susceptible varieties for the three rusts.

#### 2.1.2. Sampling

Samples were collected and kept in paper bags (6 × 20 cm). For each sample taken, all relevant information was recorded, including the date, governorate, cultivar, severity, and the collector’s name. Collected samples (infected leaves) were preserved in an icebox, then left in their bags overnight at room temperature to reduce the humidity in the samples. Samples then were preserved in desiccators containing calcium chloride under 4 **°**C in a refrigerator [29].

#### 2.1.3. Isolation and Purification

Samples with enough urediniospores were utilized to inoculate seedlings of the highly susceptible wheat variety “Morocco” at 7 days old. The seedlings were treated by spraying with water that contained a few drops of Tween-20 before being inoculated with a spatula. The infected plants were maintained in a growth chamber for 24 h at 100% humidity [29]. The plants were transferred to greenhouse benches to produce rust pustules. To collect sufficient urediniospores for inoculating the differential sets (*Lr*s), three single pustules were collected separately from each sample and used to inoculate the seedlings of a highly susceptible wheat variety (Morocco).

#### 2.1.4. Race Identification

The procedure for identifying leaf rust races was based on inoculating isogenic lines (*Lr*s) with urediniospores of *P. triticina* [30]. The plant response was determined using this method on twenty lines separated into five sets of four lines. The first set included isogenic Lr lines *1*, *2a*, *2c* and *3*; the second—*9*, *16*, *24* and *26*; the third group—*3Ka*, *11*, *17* and *30*; the fourth—*10*, *18*, *21* and *2b*, and the fifth—*Lr14b*, *Lr15*, *Lr36* and *Lr42*. Plants representing each rust agent isolate were labeled in letters based on a mixture of low infection type (L) and high infection type (H) responses. As a consequence, each pathotype is assigned a code consisting of five consonant letters from the English alphabet, ranging from B to T (Appendix A).

### 2.2. Disease Assessment

At the seedling stage, the infection types for all near-isogenic lines were recorded at 12 days after the formation of pustules on near-isogenic lines, using a disease rating scale of 0 to 4 (Appendix A) [31]. Based on the infection types developed by each line, the virulence patterns on different sets were assessed. Low infection was represented by infection types 0, 1, and 2, while high infection was represented by infection types 3 and 4 [30].

At the adult stage, disease severity (DS%) was reported for the four governorates on 16 wheat cultivars, based on the percentage of leaf area covered with rust pustules. Field reaction of leaf rust infection types was classified into five categories: highly resistant (0), resistant (R), moderately resistant (MR), moderately susceptible (MS), and susceptible (S) [29].

### 2.3. RAPD Analysis of DNA for P. triticina Races

#### DNA Extraction and PCR Amplification

Genomic isolation was used to extract DNA from spores (17 races) according to the technique of Dellaporta et al. [32]. In a 25 μL reaction volume, a polymerase chain reaction was carried out using 2.5 μL of 50 ng/μL genomic DNA, 1 μL each primer (10 pmol, F&R), and 8 μL MQ H_2_O [33]. Electrophoresis of the amplification products was done at 100 V/1 h. The gel was stained with ethidium bromide after electrophoresis, and bands were seen with UV light and photographed using a Syngen UV visualizer. The tested primers’ nucleotide sequence, and G + C percentage (Bioneer Company, Oakland, CA, USA) were utilized in RAPD experiment (Appendix A).

### 2.4. Efficacy of Resistant Genes to Wheat Leaf Rust at Seedling Stage

Evaluation of 22 monogenic lines (*Lr*s) presented in Appendix A against physiological races was conducted in the greenhouse of Wheat Diseases Dept., PPRI, ARC, Giza. Inoculation, incubation, and disease assessment were carried out as previously mentioned.

To evaluate leaf rust resistance genes, it is better to consider the number of susceptible responses over the total number of responses to the isolates. However, virulence percent was determined by recording the number of resistant responses over the total responses, as described by Green [34], as follow:$$Gene\;efficacy\;\left(\%\right)=\frac{No.\;of\;avirulent\;isolates}{Total\;number\;of\;isolates}\times 100\;Virulence\;\left(\%\right)=\frac{No.\;of\;virulent\;isolates}{Total\;number\;of\;isolates}\times 100$$

### 2.5. Molecular Markers Procedure for Identification of Resistance Genes

#### 2.5.1. Plant Sampling

Sixteen Egyptian wheat cultivars, i.e., Gemmeiza-7, Gemmeiza-9, Gemmeiza-10, Gemmeiza-11, Gemmeiza-12, Sakha-93, Sakha-94, Sakha-95, Sids-1, Sids-12, Sids-13, Sids-14, Misr-1, Misr-2, Misr-3 and Shandweel-1, as well as two monogenic lines *(Lr19* and *Lr28*) were used for identifying leaf rust resistance gene(s).

#### 2.5.2. Polymerase Chain Reaction

DNA was extracted following the procedures of Dellaporta et al. [32]. PCR was carried out with a 2.5 μL reaction volume comprising 2.5 μL of 50 ng/L genomic DNA, 1 μL of forward and reverse primers (10 pmol), and 8 μL of MQ H_2_O [33]. Appendix A shows the unique SSR primers used to confirm the expression of the *Lr19* and *Lr28* genes in 16 cultivars.

PCR products (20 μL) were electroporated on 1.5% agarose gel in 1× TBE buffer. DNA Ladder Plus (100 bp-3 kbp, Jena Bioscience, M-203) was utilized to measure the size of DNA fragments. Ethidium bromide was used to stain the gel, then photographed with a Syngen™ UV Transilluminator.

### 2.6. The Principal Component Analysis (PCA) and Two-Way Hierarchical Cluster Analysis (HCA)

Principal component analysis (PCA) was performed using the number of isolates and frequency (%) of *P. triticina* races and the associated loading plots were generated using the singular value decomposition (SVD). In addition, standardized two-way hierarchical cluster analysis (HCA) was used to differentiate the interactions between the 17 individual physiological races of *P. triticina*, 16 Egyptian commercial wheat cultivars, and 4 main wheat-growing governorates, based on the number of isolates, frequency (%), and virulence formula of *P. triticina* races in these populations. Distance and linkage were done using Ward’s minimum variance method [35] with 95% confidence between groups. The differences in the isolates number and frequency are also visualized and presented as a heat map, combined with two-way HCA, as described above.

## 3. Results

Overall, we sampled 179 wheat samples that show the characteristic symptoms of leaf rust disease from 16 different cultivars grown in four different locations during the 2019/2020 and 2020/2021 growing seasons, which yielded approximately 130 isolates (Table 1). The first season (2019/2020) yielded the greatest number of samples (94 out of 179 samples) and isolates (71 out of 130 isolates) (Table 1). In terms of locations, the Sharkia governorate had the highest number of samples (27 and 25 samples during 2019/2020 and 2020/2021, respectively) and isolates (20 and 17 isolates during 2019/2020 and 2020/2021, respectively), followed by Kafrelsheikh and Beheira, while the lowest numbers of samples and isolates were recorded from Alexandria (Table 1).

### 3.1. Assessment of Wheat Cultivars Resistance to Leaf Rust at Adult Stage

Disease severity (%) of wheat leaf rust on 16 Egyptian cultivars at four different governorates, included Kafrelsheikh, Beheira, Sharkia, and Alexandria were evaluated during the 2019/2020 and 2020/2021 growing seasons. In both seasons, Gemmeiza-12 and Misr-3 cultivars were resistant at all studied locations (Table 2 and Figure 1A). However, all other cultivars showed different degrees of susceptibility to *P. triticina* in most studied locations, particularly new cultivars, such as Sakha-95, Sids-14, and Shandweel-1 (Table 2 and Figure 1A). Gemmeiza-7, Gemmeiza-11, Sakha-93 and Sids-1 were the most susceptible wheat cultivars in the four governorates during the two seasons (Figure 1A). In terms of locations, the Sharkia governorate recorded the highest number of susceptible wheat cultivars in both seasons, followed by Alexandria, then Beheira and Kafrelsheikh, which were almost similar, particularly during the 2019/2020 season (Figure 1B).

### 3.2. Distribution of P. triticina Isolates between Different Cultivars, Locations, and Physiological Races

During the 2019/2020 season, the highest number of isolates was recorded for Gemmeiza-7 and Gemmeiza-11, followed by Sids-1 and Sakha-93, respectively, whereas during the 2020/2021 season, the highest number of isolates was recorded for Gemmeiza-11, followed by Sakha-93, Sids-1, and Gemmeiza-7, respectively (Figure 1C). In both seasons, the highest number of isolates was recorded on samples collected from the Sharkia governorate, followed by Kafrelsheikh, Beheira, and Alexandria, respectively (Figure 1D).

### 3.3. The Effect of Location-Cultivars Association on the Number of Isolates and Their Frequency (%)

Generally, 130 isolates were identified from 16 different cultivars grown in 4 different locations during the two growing seasons. However, a higher number of *P. triticina* isolates was recorded during the 2019/2020 (71 isolates) than the 2020/2021 season (59 isolates) (Table 3). In both seasons, the highest number of isolates was recorded in Sharkia (20 and 17 isolates during 2019/2020 and 2020/2021, respectively), followed by Kafrelsheikh (19 and 15 isolates during 2019/2020 and 2020/2021, respectively) and Beheira (17 and 14 isolates during 2019/2020 and 2020/2021, respectively), whereas the lowest number of isolates was recorded in Alexandria (15 and 13 isolates during 2019/2020 and 2020/2021, respectively) (Table 3).

In terms of cultivars, in the 2019/2020 season, the highest numbers of isolates were identified from Gemmeiza-7 and Gemmeiza-11 (11 isolates that donated approximately 15.5% frequency). While in the 2020/2021 season, Gemmeiza-11 recorded the highest number of isolates (9 isolates that donated approximately 15.3% frequency), followed by Sakha-93 and Sids-1, which recorded eight isolates each (about 13.6 frequency). Interestingly, no disease symptoms (pustules) were recorded for Gemmeiza-12 and Misr-3 at the four studied governorates during both seasons, which support the suggestion that both cultivars are resistant to *P. triticina* (Table 3).

### 3.4. PCA Reveals Differences in Wheat Cultivation among Different Locations

Principal component analysis (PCA) was performed using the number of isolates of individual cultivars in the four studied locations. The PCA-associated scatter plots and loading plots are shown in Figure 2A,B. The scatter plot obtained using the 2019/2020 data showed a clear separation among all studied locations/governorates (Figure 2A). However, in 2020/2021, both the Sharkia and Alexandria governorates were overlapped each other slightly in the upper-left side of the scatter plot (Figure 2B). In the 2019/2020 season, PC1 and PC2 were responsible for 95.47% of the variation, whereas in the 2020/2021 season they were responsible for only 76.12% of the variation. Moreover, the loading plot showed that the isolates collected during 2019/2020 from six cultivars (Misr-2, Gemmeiza-9, Sids-14, Sakha-95, Sakha-94, and Misr-1) were positively correlated with the Sharkia governorate and five cultivars (Sids-1, Gemmeiza-10, Shandweel-1, Gemmeiza-11, and Gemmeiza-7) were correlated with the Kafrelsheikh governorate. Nevertheless, only two cultivars were correlated with the Beheira governorate (Sids-12 and Sakha-93) and Sids-13 was correlated with the Alexandria governorate (Figure 2A). On the other hand, in the 2020/2021 season, isolates collected from only four cultivars (Sakha-94, Misr-1, Gemmeiza-9, and Sakha-95) were correlated with the Sharkia and Alexandria governorates and three cultivars (Gemmeiza-7, Sids-12, and Sakha-93) were correlated with the Kafrelsheikh governorate, while only two cultivars were correlated with the Beheira governorate (Gemmeiza-10 and Gemmeiza-11) (Figure 2B).

Additionally, standardized two-way hierarchical cluster analysis (HCA) was used to differentiate the individual cultivars among the studied location based on the number of isolates and frequency (Figure 2C,D). The differences in the isolates’ number and frequency are also visualized and presented as a heat map. In both seasons, the total HCA dendrogram among locations/governorates (presented at the bottom of Figure 2C,D) showed that the cultivar profiles of Kafrelsheikh were closer to the profile of Beheira, whereas that of Sharkia was closer to that of Alexandria.

### 3.5. Race Analysis of P. triticina Isolates

To evaluate the range of pathogenic variation in a particular area, the race analysis was carried out for all isolates based on the reaction of distinct lines containing 20 monogenic resistance genes at the seedling stage in the greenhouse. Briefly, 17 physiological races were identified during the seasons 2019/2020 and 2020/2021 (Table 4). The pathological pathotype MTTGT was the most abundant race (21 isolates in both seasons, which donated about 16.20% of total frequency), followed by STSJT (19 isolates), TTTKS (17 isolates), and TTTST (16 isolates) as a total of both seasons, while MBGJT was the lowest abundant race (only one isolate during 2019/2020 season) (Table 4). It worth noting that, both pathological pathotypes, MBGJT and PKTTS, appeared in only the first season and disappeared in the second season.

### 3.6. PCA Reveals Differences in the Distribution of Physiological Races of P. triticina among Locations

PCA was performed using the number of isolates of individual physiological races of *P. triticina* in the four studied locations (Figure 3A,B). Generally, in both seasons, the scatter plots showed a clear separation among all studied locations/governorates (Figure 3A,B). PC1 and PC2 were responsible for 93.15% and 92.15% of the variation during the 2019/2020 season and 2020/2021 season, respectively. Additionally, the loading plot using the 2019/2020 data showed that seven physiological races (PTKKT, PKTTS, PKKSR, STTLT, NRNJT, LTTFT, and TTTST) were positively correlated with the Alexandria governorate and six pathological pathotypes (MBGJT, LTCGT, PTTJT, STSJT, TTTKS, and PBPPP) were correlated with Beheira governorate (Figure 3A). Though, only two pathotypes were associated with Sharkia (MTTGT and GTTPT) and Kafrelsheikh governorate (LGDTT and HTTDS) (Figure 3A).

On the other hand, during the 2020/2021 season, five physiological races (NRNJT, TTTST, HTTDS, PKKSR, and LTTFT) were correlated with the Alexandria governorate. Likewise, five pathological pathotypes (GTTPT, PTTJT, TTTKS, LTCGT, and STSJT) were correlated with the Kafrelsheikh governorate. However, only three physiological races (PBPPP, LGDTT, and STTLT) were correlated with the Beheira governorate and two pathological pathotypes (MTTGT and PTKKT) were associated with the Sharkia governorate (Figure 3B).

Furthermore, the standardized two-way HCA was used to differentiate the interactions between the 17 individual physiological races of *P. triticina* and various studied governorates, based on the number of isolates, frequency (%), and virulence formula of *P. triticina* races in these populations (Figure 3C,D). The differences in the isolates number and frequency are also visualized and presented as a heat map. In both seasons, the total HCA dendrogram among locations/governorates (presented in the bottom of Figure 3C,D) showed that the distribution of physiological races of *P. triticina* in Kafrelsheikh was closer to the profile of Beheira, whereas the distribution of physiological races of *P. triticina* in Sharkia was closer to that of Alexandria.

During the 2019/2020 season, the total HCA dendrogram among physiological races of *P. triticina* (presented on the left side of Figure 3C) showed that the 17 identified races were separated into four distinct clusters. The first cluster (C-I) included four pathotypes (GTTPT, PKTTS, PTKKT, and TTTST), which were dominant in the Sharkia and Alexandria governorates. Likewise, the second cluster (C-II) included four pathotypes (HTTDS, LGDTT, PBPPP, and PTTJT), which were mainly prominent in the Kafrelsheikh governorate and some in the Beheira governorate (Figure 3C). The third cluster (C-III) included six pathotypes (LTCGT, MBGJT, LTTFT, NRNJT, PKKSR, and STTLT). The last cluster (C-IV) included the three most frequent physiological races of *P. triticina* (MTTGT, STSJT, and TTTKS) that appeared in almost all studied locations.

It worth mentioning that the total HCA dendrogram of the physiological races of *P. triticina* during the 2020/2021 season (presented on the left side of Figure 3D) was slightly different to the one described above. Briefly, the 17 physiological races were separated into six distinct clusters. C-I included four pathotypes (GTTPT, LTCGT, PTTJT, and STTLT), C-II included five pathotypes (HTTDS, LTTFT, PKKSR, NRNJT, and PTKKT), C-III included only two pathotypes (LGDTT and PBPPP), C-IV included only two pathotypes (MBGJT and PKTTS), C-V included only two pathotypes (STSJT and TTTKS), and C-VI included only two pathotypes (MTTGT and TTTST) as well (Figure 3D) which were dominant in Sharkia and Alexandria governorates. Collectively, these findings suggest that the most virulent race (TTTST) in the Sharkia and Alexandria governorates infected the new wheat cultivars; Sakha-94, Sakha-95, and Shandweel-1.

### 3.7. Identification of Physiological Leaf Rust Races on the Egyptian Wheat Cultivars

Due to the potential of *P. triticina* to develop new races that can target resistant cultivars under favorable environmental conditions, the association between physiological races of *P. triticina* and the Egyptian wheat cultivars was deeply studied and presented in Table 5. In the Kafrelsheikh governorate, mainly, five main pathotypes (GTTPT, HTTDS, PTTJT, STSJT, and TTTKS) were identified in the six wheat cultivars (Gemmeiza-7, Gemmeiza-11, Sakha-93, Sids-1, Sids-12, and Shandweel-1) during both seasons. In addition, the physiological races LGDTT and PBPPP were identified only during the 2019/2020 season, and not 2020/2021, for the Gemmeiza-10 and Sakha-93 cultivars, respectively. Likewise, the physiological races LTCGT and STTLT were identified only during the 2020/2021 season, but not 2019/2020, on Sids-1 and Sids-14 cultivars, respectively (Table 5).

In the Beheira governorate, generally, six main physiological races (GTTPT, LTCGT, PBPPP, PTTJT, STSJT, and TTTKS) were identified on six wheat cultivars (Gemmeiza-7, Gemmeiza-11, Sakha-93, Sids-1, Sids-12, and Shandweel-1) during both seasons. In addition to these pathotypes, MTTGT and MBGJT were identified only during the 2019/2020 season, but not 2020/2021, for Gemmeiza-7 and Gemmeiza-10, respectively. Similarly, the LGDTT and STTLT pathotypes were identified only during the 2020/2021 season, but not in 2019/2020, for Gemmeiza-10, Sids-13 and Sids-14 cultivars, respectively (Table 5).

In the Sharkia governorate, five main physiological races (GTTPT, MTTGT, PTKKT, STTLT, and TTTST) were largely identified for all studied wheat cultivars, except Gemmeiza-12 and Misr-3, which appeared to be resistant to all identified physiological races of *P. triticina* in the four studied governorates during both seasons. In addition, three pathotypes (PKTTS, PKKSR, and TTTKS) were only identified during the 2019/2020 season, and not 2020/2021 (Sids-1, Sids-13, and Sids-14, respectively). Moreover, a further three pathotypes (PTTJT, HTTDS, and NRNJT) were identified only during the 2020/2021 season, and not 2019/2020, for Sakha-93, Sids-1, and Sids-13 cultivars, respectively (Table 5).

Finally, in the Alexandria governorate, six main pathotypes (LTTFT, MTTGT, NRNJT, PKKSR, STTLT, and TTTST) were primarily identified for ten wheat cultivars (Gemmeiza-7, Gemmeiza-11, Sakha-93, Sakha-94, Sakha-95, Sids-1, Sids-12, Sids-13, Sids-14, and Shandweel-1) during both seasons. Additionally, five physiological races (STSJT, PTKKT, PKTTS, TTTKS, and GTTPT) were identified only during the 2019/2020 season, but not 2020/2021, on Gemmeiza-7, Sakha-93, Sids-1, Sids-14, and Misr-1 cultivars, respectively. Likewise, the physiological race HTTDS was identified only during the 2020/2021 season, but not 2019/2020, on the Gemmeiza-7 cultivar (Table 5).

In the 2019/2020 season, the most frequent race, MTTGT (Table 5), was recorded in the Sharkia, Beheira, and Alexandria governorates, but not Kafrelsheikh. However, this race was reported only in Sharkia and Alexandria, but neither the Beheira nor Kafrelsheikh governorates during the 2020/2021 season (Table 5). Besides, the second frequent race, STSJT, was recorded in Kafrelsheikh, Beheira, and Alexandria, but not Sharkia during the 2019/2020 season. Nevertheless, during the 2020/2021 season, STSJT was only reported in samples collected from Kafrelsheikh and Beheira, but neither Alexandria nor Sharkia (Table 5). Moreover, the third most frequent race, TTTKS, was identified from all studied locations during the 2019/2020 season; however, it was reported only from Kafrelsheikh and Beheira, but neither the Alexandria nor Sharkia governorates during the 2020/2021 season (Table 5).

### 3.8. PCA Reveals Differences in the Distribution of Physiological Races of P. triticina among Cultivars

PCA was performed using the number of isolates of individual physiological races of *P. triticina* on the 16 studied cultivars (Figure 4A,B). Briefly, in both seasons, scatter plots showed an accepted separation among all studied cultivars (Figure 4A,B). PC1 and PC2 were responsible for 65.99% and 58.49% of the variation during the 2019/2020 and 2020/2021 seasons, respectively.

It worth mentioning that Gemmeiza-9, Gemmeiza-10, Gemmeiza-12, Misr-1, Misr-2, Misr-3, Sids-13, Sakha-94, and Sakha-95 were clustered together in the negative quarter of the scatter plot during the 2019/2020 season (Figure 4A). However, only Gemmeiza-12, Misr-1, Misr-2, and Misr-3 were clustered together in the same part of the graph during the 2020/2021 season (Figure 4B).

Additionally, the loading plot using 2019/2020 data showed that the most frequent/abundant physiological races MTTGT and STSJT strongly influenced PC1, whereas the TTTKS pathotype had more effect in PC2. Moreover, the physiological races PTKKT, PBPPP, PTTJT, PKKSR, GTTPT, NRNJT, MBGJT, LTTFT, and LGDTT were clustered together and appeared to have less weight on PC1 and PC2 (Figure 4A). On the other hand, during the 2020/2021 season, the pathotypes TTTKS and TTTST strongly affected PC1, while MTTGT, STSJT, and STTLT strongly manipulated PC2 (Figure 4B).

Furthermore, standardized two-way HCA was used to differentiate the individual physiological races of *P. triticina* among the studied cultivars based on the number of isolates and frequency (Figure 5A,B). The differences in the isolates’ number and frequency are also visualized and are presented as a heat map. In both seasons, Gemmeiza-7, Sids-12, and Sids-1 were clustered together (C-I) which appeared to have a higher frequency of the physiological races MTTGT and STSJT (Figure 5A,B). However, Sakha-93 was also clustered with C-I during the 2020/2021 season (Figure 5B). Likewise, Gemmeiza-9, Sakha-94, and Gemmeiza-10 were clustered together, within C-II, which appeared to be associated with the pathotype TTTST in both seasons. Nevertheless, Sakha-95 was also clustered with C-II during the 2019/2020 season (Figure 5B). Additionally, the resistant cultivars Gemmeiza-12 and Misr-3 were cluster together with C-III during both seasons.

### 3.9. Random Amplified Polymorphic DNA (RAPD) Assay Cluster Analysis Using RAPD Markers

The races were split into two major clusters (Figure 6 and Figure 7), the first of which was further divided into two sub-clusters. At 300 bp, the first sub-cluster contained the races GTTPT, MTTGT, STTLT, TTTKS, and TTTST, while the second sub-cluster included PKTTS, PTKKT, PTTJT and STSJT at 150 bp. HTTDS, LGDTT, LTCGT, LTTFT, MBGJT, NRNJT, PBPPP, and PKKSR were part of the second cluster, which was split into one more sub-cluster at 200 bp.

Since it captures all variation identified by each primer, cluster analysis utilizing all three primers certainly has greater power in separating the investigated isolates, giving a more meaningful grouping pattern.

### 3.10. Effectiveness of Leaf Rust Resistance Genes at Seedling Stage

For each line, the frequency of virulence was calculated as the ratio of virulent cultures to the total number of cultures. The frequency of virulence to *Lr* genes varied between *P. triticina* regional populations in Egypt. Data presented in Table 6 show different frequencies of virulence to the tested *Lr* genes. The least frequencies of virulence were found in *Lr19, Lr28, Lr2b,* and *Lr10* showing 0.00, 0.00, 39.44, and 38.03%, respectively, in the first season and 0.00, 0.00, 23.73, and 44.07%, respectively, in the second season. On the other hand, virulence against *Lr1*, *Lr2a*, *Lr2c*, *Lr3*, *Lr3ka*, *Lr9*, *Lr11*, *Lr14b*, *Lr15*, *Lr16*, *Lr17*, *Lr18*, *Lr21*, *Lr24*, *Lr26*, *Lr30*, *Lr36*, and *Lr42* showed the highest frequencies over 57.69%. It worth noting that *Lr19* and *Lr28* exhibited complete resistance to leaf rust in the current study under Egyptian field conditions (Table 6).

### 3.11. Molecular Markers

Sixteen wheat cultivars and two resistant lines (*Lr* genes) *Lr19* and *Lr28* were selected for the identification of resistance genes using molecular markers. Briefly, two specific primers were used to detect two resistance genes (Figure 8). The markers for *Lr19* and *Lr28* were defined as 150 bp and 300 bp fragments, respectively, in two wheat cultivars (Gemmeiza-12, and Misr-3) during the polymorphic survey, while fourteen of the tested cultivars did not show the presence of *Lr19* (Figure 8A) and *Lr28* (Figure 8B). These findings might explain the molecular mechanism(s) behind the previous results that the two cultivars, Gemmeiza-12 and Misr-3, are resistant to all races of leaf rust pathogen.

## 4. Discussion

Wheat leaf rust was responsible for the extinction of numerous cultivars in Egypt, including Giza-139, Mexipak-69, Super-X, and Chenab-70, due to their susceptibility under field conditions. Moreover, some wheat genotypes were discarded very shortly after their release, such as Giza-139 [36]. The failure of these cultivars was attributed to the broad pathogen populations, as well as their strong evolutionary potential. The pathogen’s dynamic nature resulted in the quick production and emergence of several new virulent races with the capacity to overcome resistance genes in wheat cultivars. As a consequence, the effective lifespan of these recently released cultivars is constantly reduced [11,12,16]. Therefore, the identification of the dominant physiological races in each area is one of the most important steps in rust-resistance breeding programs. The program will be effective if all physiological isolates of a pathogen are utilized [37]. Generally, new leaf rust races emerge as a result of mutation [38], heterokaryosis [39], gene recombination [40,41], migration [42], vegetative or parasexual recombination, hyphal anastomosis [43] and natural selection of the most virulent races in the region [44].

The alternative host (*Thalictrum* spp.) was not discovered in Egypt, and the urediniospores are unable to live in Egypt during the summer owing to the high temperatures [14]. The initial inoculum is frequently transported by the prevailing winds from neighboring countries. Therefore, the current research focused on the frequency of various virulence to investigate changes in virulence formula throughout two seasons. During the growing seasons 2019/2020 and 2020/2021 in Egypt, a survey for wheat leaf rust revealed the prevalence of the disease caused by *P. triticina* in four governorates, i.e., Kafrelsheikh, Beheira, Sharkia, and Alexandria. Most of the infected samples were collected from the commercial fields and the Egyptian wheat trap rust nurseries (EWTRN). Wheat infected samples (around 179 samples) were collected from 16 distinct cultivars growing in four different locations for the two annual surveys (2019/2020 and 2020/2021) yielding roughly 130 isolates. During the two seasons under investigation, Sharkia governorate had the largest number of samples, isolates, and susceptible wheat cultivars, followed by Kafrelsheikh and Beheira, while Alexandria had the lowest number of samples and isolates. In addition, the total HCA dendrogram across locations/governorates in both seasons revealed that the cultivars profile of Kafrelsheikh was closer to that of Beheira, whilst the profile of Sharkia was closer to that of Alexandria. This is because wheat is more often grown in these governorates and the environmental conditions are more suitable for the spread of the disease. Thus, these governorates were regarded as leaf rust hotspots [11,17].

Gemmeiza-12 and Misr-3 cultivars were resistant in all tested locations over the two seasons, with no disease symptoms (pustules) observed in these cultivars in the four governorates. Additionally, the cultivars Gemmeiza-12 and Misr-3 were clustered together during both seasons. However, the new wheat cultivars, such as Sakha-95, Sids-14, and Shandweel-1, were susceptible to *P. triticina* in most of the studied locations. This could be because the most common leaf rust pathotypes change from year to year in pathogen populations [18,45]. Therefore, it can be suggested that wheat production in Egypt is highly correlated with the changes in cultivar distribution to enhance yield and resistance to fungal infections in new cultivars. Wheat cultivation increased, and the distribution of the most commonly planted cultivars varied considerably as well. Generally, more than half of the wheat cultivated area was planted with seven cultivars: Gemmeiza-11, Sakha-94, Sids-12, Sids-14, Misr-1, Misr-2, and Shandweel-1. However, they are now susceptible or moderately susceptible to wheat leaf rust [46]. Hence, it was necessary to identify the physiological races that supplant the resistance in wheat cultivars. We would like to know whether the physiological races reported in the same governorates may be found in the same cultivars in different governorates. This motivated us to investigate the virulence dynamics, diversity, and degree of similarity of *P. triticina* populations in various areas, as one of the most crucial stages in rust resistance breeding programs [47].

Race analysis was carried out in the greenhouse by recording the response of different lines containing 20 monogenic resistance genes at the seedling stage. These genes have varied reactions in different race groups, so they are race-specific. Race analysis was important for determining the range of pathogenic variation in a particular area. The screening for resistance in cultivars revealed that the race differences are responsible for host responses, contributing to our understanding of the mechanism of diversity and driving research and breeding programs to avoid future disease outbreaks.

Seventeen physiological races were identified during the seasons 2019/2020 and 2020/2021. The pathotype MTTGT was found in the Sharkia, Beheira, and Alexandria governorates and was the most common race in both seasons. The second race, STSJT, was found in Kafrelsheikh and Beheira for Gemmeiza-7, Sids-1, and Sids-12 cultivars, as well as in the Alexandria governorate on Gemmeiza-7 cultivars. Moreover, TTTST was the most virulent race in the Sharkia and Alexandria governorates and disappeared in other governorates. The standardized two-way HCA dendrogram between locations/governorate revealed that the distribution of physiological races of *P. triticina* in Kafrelsheikh was closer to the profile of Beheira, while that of the physiological races of *P. triticina* in Sharkia was closer to the profile of Alexandria. Additionally, only two pathotypes (MTTGT and TTTST) were detected in the same cluster in the Sharkia and Alexandria governorates. These data imply that the most virulent race (TTTST) infected the new wheat cultivars Sakha-94, Sakha-95, and Shandweel-1 in the Sharkia and Alexandria governorates.

Both pathotypes, MBGJT and PKTTS, originally occurred in the first season and then disappeared in the second. The long-distance dispersal of pathogen propagules and the high gene/genotype flow of such pathogens were mainly held responsible for changes in wheat leaf rust populations and the appearance or disappearance of virulence [44]. This process resulted in noticeable changes in the genetic structure of the recipient pathogen populations, particularly during the summer. Foreign sources of leaf rust inoculum enter Egypt every year, and it is transported from one region to another within the same year [15]. Furthermore, previous studies in Egypt confirmed the conclusion that the source of primary inoculum, which is often carried in by northern winds each year from foreign sources, determines the presence of virulent races in leaf rust populations [14]. According to recent research, urediniospores of leaf rust do not persist or cannot survive in Egypt during the summer [12]. The current study was supported by the results of McVey et al. [15] and Negm et al. [16] who suggested that leaf rust populations in Egypt are made up of a wide range of races.

The races prevailed in certain governorates and disappeared in others. This could be due to climatic differences between governorates, such as temperature, wind directions, and rainfall, which are all necessary for disease occurrence and the potential of *P. triticina* to develop new races that can attack resistant cultivars under favorable environmental conditions, resulting in significant losses [9,48]. These results are supported by previous reports indicating that changes in the virulence and genetic structure of leaf rust populations are mostly affected by some environmental factors, such as temperature and relative humidity in the wheat-growing regions [12,47].

In the current study, the differences between the races of *P. triticina* obtained from various regions in Egypt were investigated using RAPD analysis to determine their relationship. This research revealed that *P. triticina* collections vary on a global scale in terms of virulence and molecular background. *P. triticina* race analysis revealed differences in virulence and RAPD phenotypes. Previous research has shown that arbitrary decameter primers were used to examine virulence polymorphism in 20 near-isogenic wheat differential lines and three randomly amplified polymorphic DNA (RAPD) [44]. The cluster analysis revealed a correlation between virulence and molecular variation in general [44]. The results support the idea that *P. triticina* has different regional populations in Canada [44]. The most molecular diversity was found across races with varied virulence phenotypes. There were minor molecular differences across races of similar virulence, with the first cluster consisting of virulent races. The molecular polymorphisms were more efficient in discriminating between the main clusters of *P. triticina* compared to virulence polymorphisms. There was a correlation between the virulence and molecular dissimilarity matrixes. In *P. triticina* from Canada, cluster analysis revealed a link between virulence and molecular polymorphism [27]. *P. triticina* was collected from the different locations. All simple uredinial isolates from the collection were evaluated for virulence polymorphism on 22 Thatcher lines according on virulence phenotypes, and chosen isolates were further analyzed for RAPD using 11 primers [27]. There were 105 virulence phenotypes and 82 RAPD phenotypes among the 131 simple uredinial isolates. Differences in isolates across groups were responsible for 36% of the RAPD variation. RAPD analysis was able to display the connection between the races of *P. triticina*, which vary in terms of virulence and were collected from different locations in Egypt. Seventeen races were divided into three categories based on RAPD analysis: 150, 200, and 300 bps. GTTPT, MTTGT, STTLT, TTTKS, and TTTST were all members of the same cluster of pathogenic races. As a result, it’s more probable that these races (members of the same cluster) shared a lot of genetic material and came from the same location. Furthermore, the isolates that were the most closely related seemed to be from different sources. Since it captures all variation identified by individual primers, cluster analysis employing all four primers probably has greater power in distinguishing among examined isolates, and therefore provides a more meaningful grouping pattern. The results of the cluster analysis were consistent, indicating that the findings were reliable, and that RAPD markers may be used to evaluate *P. triticina* genetic diversity. Furthermore, the present research revealed that *P. triticina* has a lot of genetic variation.

As a result of the dynamic changes in leaf rust races, plant breeders should incorporate new efficient resistance genes [17,18,49]. Therefore, identifying leaf rust resistance genes is critical for adding new effective resistance genes to wheat breeding programs. During the polymorphism survey, the markers for *Lr19* and *Lr28* were identified as 150 bp and 300 bp fragments in two wheat cultivars (Gemmeiza-12 and Misr-3), while fourteen of the tested cultivars did not show the presence of *Lr19* and *Lr28*. This explains the earlier findings that both Gemmeiza-12 and Misr-3 cultivars are resistant to all races of leaf rust pathogen. Similar results were reported by Vida et al. [50], who stated that wheat genotypes containing the three-leaf rust resistance genes, *Lr9, Lr19*, and *Lr28* exhibit good and strong levels of leaf rust resistance at the adult time. In Western Siberia and the Urals, *Lr9* has been broadly utilized in breeding, but the first *Lr9*-virulent isolates were discovered in 2007 [51].

*Lr19,* which showed complete resistance to leaf rust in the current study under Egyptian field conditions is also effective in most countries in Asia, Australia, and Europe and has been related to desired genes for grain yield increase, making it a good candidate for wheat breeding [52]. Although *Lr28* has a high level of resistance to leaf rust in all common pathotypes in India, it was not related to any undesirable genes [40,53]. Therefore, it is necessary to determine if these genes are present in Egyptian wheat varieties. Therefore, the present research adds to the growing evidence that leaf rust resistance genes (R-genes) can only protect wheat crops for a limited time. On the other hand, the effect of virulent pathotypes on certain cultivars is determined by the overall level of resistance (partial resistance) in those cultivars [13,54].

## 5. Conclusions

This study is a pioneer to explain the varietal response to infection with leaf rust disease and physiological races of this fungus in Egypt. Seventeen physiological races were identified on 16 wheat cultivars, during 2019/20 and 2020/21 growing seasons by race analysis. Furthermore, using RABD analysis, the races were divided into three groups: 150, 200, and 300 bp. TTTST was the most virulent race in the Sharkia and Alexandria governorates (Figure 9) and disappeared in other governorates. It infected the new wheat cultivars in Alexandria. On the other hand, Gemmeiza-12 and Misr-3 cultivars were resistant to all races of leaf rust due to the role of resistance genes (*Lr19* and *Lr28*) in these cultivars. Finally, as a warning to the rest of the world, it must explain why some of these new races exist in Egypt. These races move to Egypt every year because there is no alternative host, and they have the potential to supplant the resistance. As a consequence, it may create problems in certain countries.

## Figures and Tables

**Figure 1 biology-10-00925-f001:**
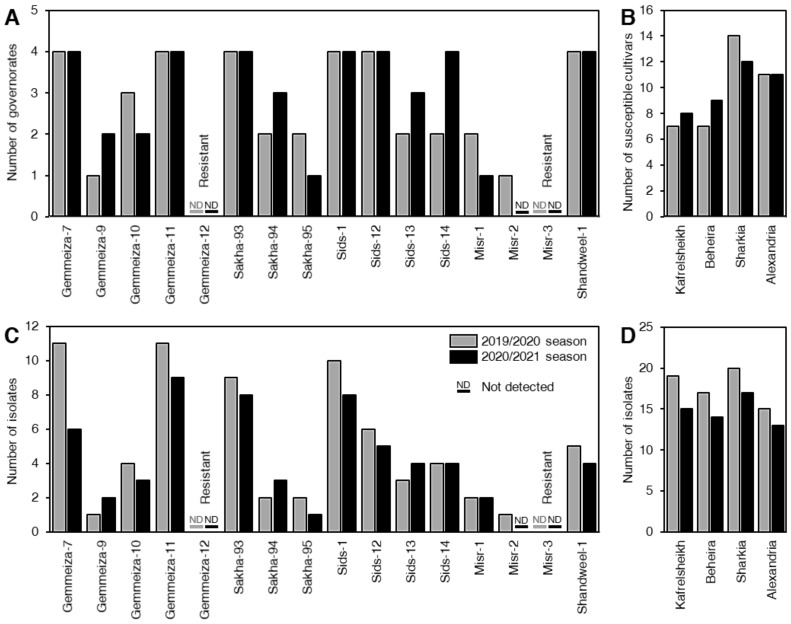
Distribution of *P. triticina* races in relation to cultivation areas and wheat cultivars. (**A**) Number of governorates that recorded susceptible wheat cultivars, (**B**) number of susceptible wheat cultivars of leaf rust in four different governorates, (**C**) number of *P. triticina* isolates recorded on 16 Egyptian commercial wheat cultivars, (**D**) number of *P. triticina* isolates recorded on each governorate.

**Figure 2 biology-10-00925-f002:**
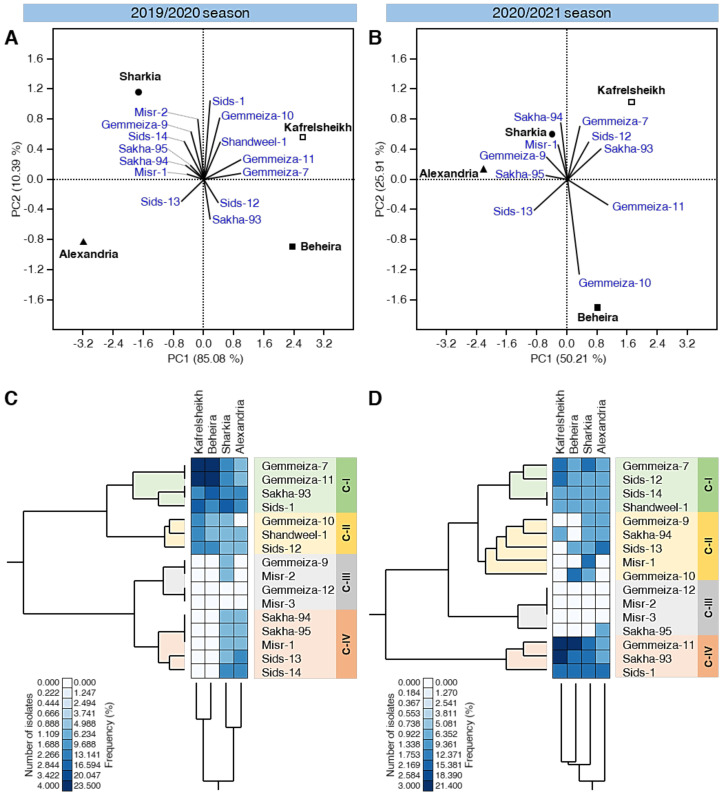
Principal component analysis (PCA) and two-way hierarchical cluster analysis (HCA) of individual commercial wheat cultivars growing in four main wheat-growing governorates in Egypt during the 2019/2020 and 2020/2021 seasons. (**A**,**B**) PCA-scatter plots and their associated loading plots using data collected during 2019/2020 and 2020/2021 seasons, respectively. (**C**,**D**) Standardized two-way HCA based on the number of isolates and frequency (%) of *P. triticina* races on 16 Egyptian commercial wheat cultivars growing in four main wheat-growing governorates in Egypt during the 2019/2020 and 2020/2021 seasons, respectively. The differences in the number of isolates and frequency (%) are visualized in the heat map diagram. Rows represent the individual cultivars, while columns represent the locations/ governorates. Low number of isolates and frequency (%) are colored light-blue and the high number of isolates and frequency (%) are colored dark-blue (see the scale at the left corner of the bottom of the heat map).

**Figure 3 biology-10-00925-f003:**
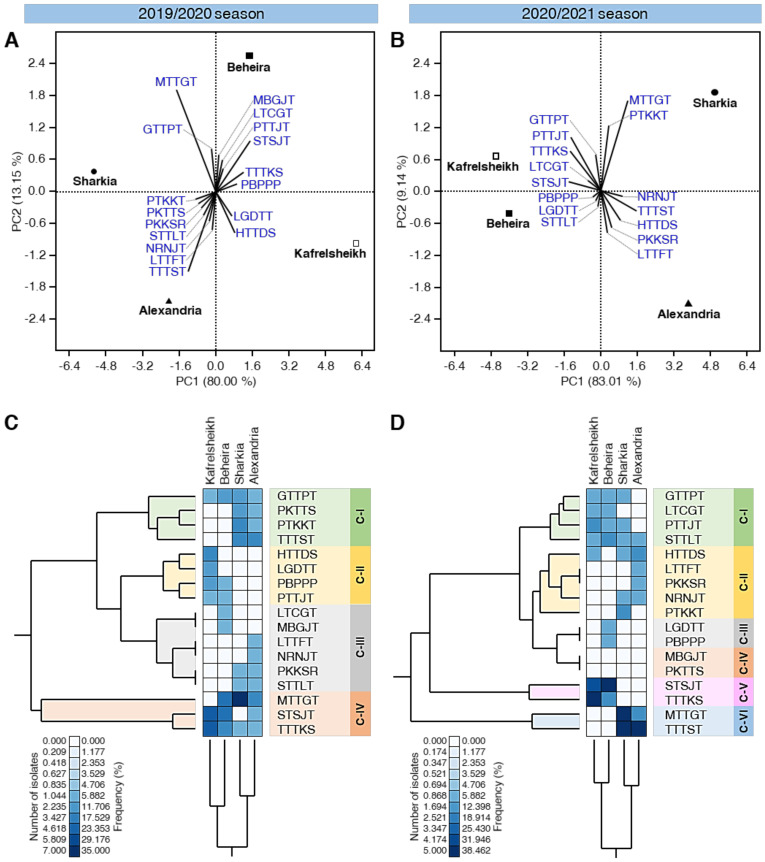
Principal component analysis (PCA) and two-way hierarchical cluster analysis (HCA) of individual physiological races of *P. triticina* detected in four main wheat-growing governorates in Egypt during the 2019/2020 and 2020/2021 seasons. (**A**,**B**) PCA-scatter plots and their associated loading plots using data collected during 2019/2020 and 2020/2021 seasons, respectively. (**C**,**D**) Standardized two-way HCA based on the number of isolates and frequency (%) of 17 individual physiological races of *P. triticina* from four main wheat-growing governorates in Egypt during 2019/2020 and 2020/2021 seasons, respectively. The differences in the number of isolates and frequency (%) are visualized in the heat map diagram. Rows represent the individual physiological races, while columns represent the locations/ governorates. Low number of isolates and frequency (%) are colored light-blue and the high number of isolates and frequency (%) are colored dark-blue (see the scale at the left corner of the bottom of the heat map).

**Figure 4 biology-10-00925-f004:**
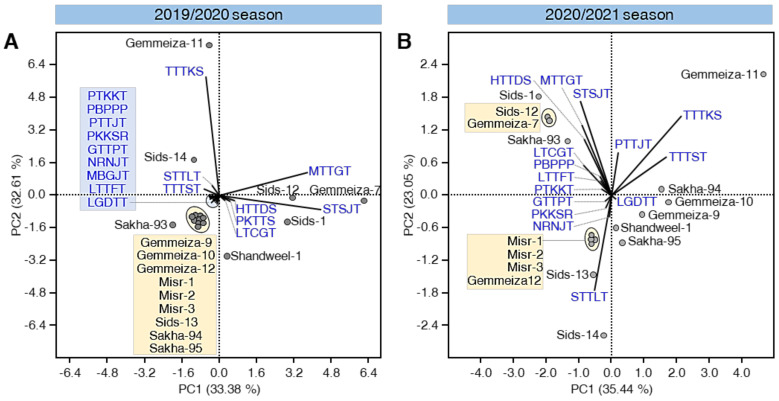
Principal component analysis (PCA) of individual physiological races of *P. triticina* detected on commercial wheat cultivars in Egypt during 2019/2020 and 2020/2021 seasons. (**A**,**B**) PCA-scatter plots and their associated loading plots using data collected during 2019/2020 and 2020/2021 seasons, respectively.

**Figure 5 biology-10-00925-f005:**
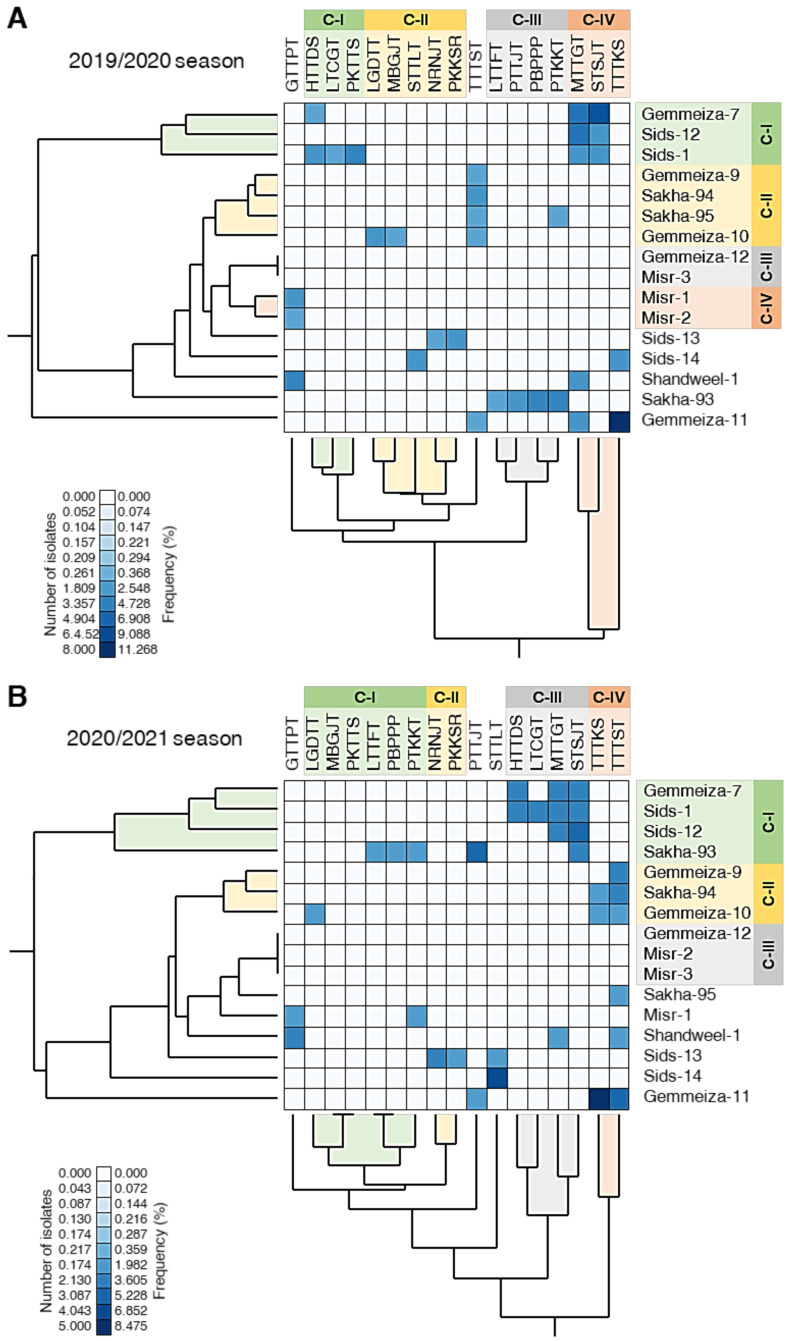
Two-way hierarchical cluster analysis (HCA) of individual physiological races of *P. triticina* detected on commercial wheat cultivars in Egypt during 2019/2020 and 2020/2021 seasons. (**A**,**B**) Standardized two-way HCA based on the number of isolates and frequency (%) of 17 individual physiological races of *P. triticina* identified from 16 commercial wheat cultivars in Egypt during 2019/2020 and 2020/2021 seasons, respectively. The differences in the number of isolates and frequency (%) are visualized in the heat map diagram. Rows represent the individual cultivars while columns represent the physiological races. Low number of isolates and frequency (%) are colored light-blue and the high number of isolates and frequency (%) are colored dark-blue (see the scale at the left corner of the bottom of the heat map).

**Figure 6 biology-10-00925-f006:**
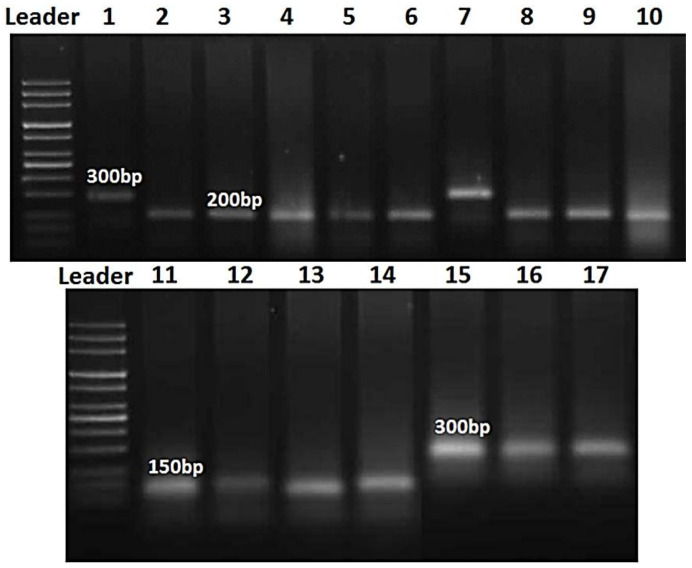
PCR electrophoretic amplification of 17 races of *Puccinia tritinia* genomic DNA. Lane 1 = GTTPT, lane 2 = HTTDS, lane 3 = LGDTT, lane 4 = LTCGT, lane 5 = LTTFT, lane 6 = MBGJT, lane 7 = MTTGT, lane 8 = NRNJT, lane 9 = PBPPP, lane 10 = PKKSR, lane 11 = PKTTS, lane 12 = PTKKT, lane 13 = PTTJT, lane 14 = STSJT, lane 15 = STTLT, lane 16 = TTTKS and lane 17 = TTTST.

**Figure 7 biology-10-00925-f007:**
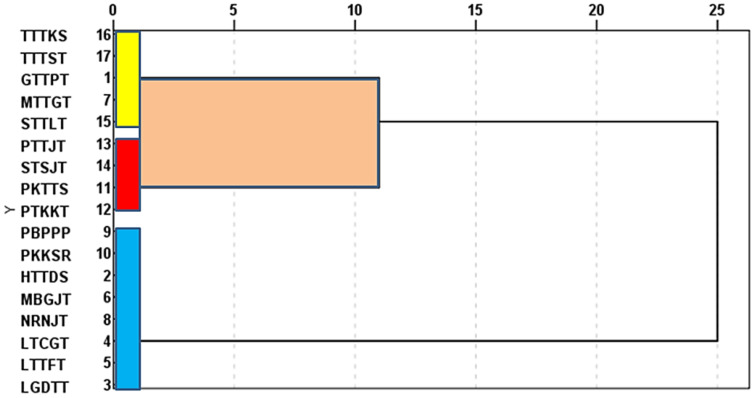
Dendrogram of 17 *Puccinia triticina* races based on the similarity in molecular weight.

**Figure 8 biology-10-00925-f008:**
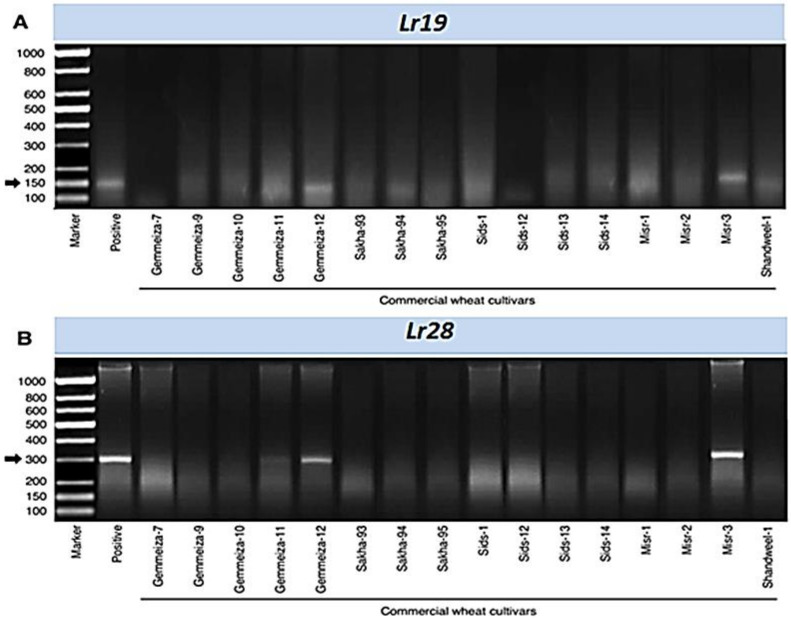
PCR electrophoretic amplification of 16 commercial wheat cultivars genomic DNA using two molecular markers. (**A**) Electrophoretic amplified patterns using the specific primer of *Lr19*. (**B**) Electrophoretic amplified patterns using the specific primer of *Lr28*.

**Figure 9 biology-10-00925-f009:**
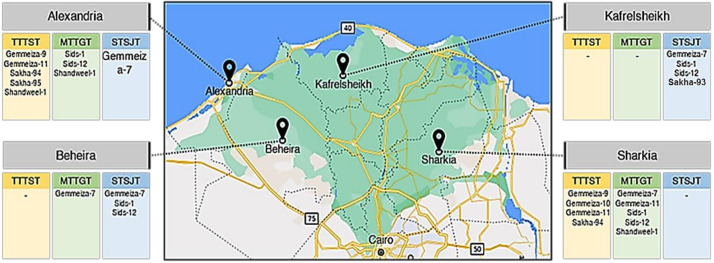
Map of distribution of the most virulent races and frequency (%) on Egyptian wheat cultivars in four different governorates.

**Table 1 biology-10-00925-t001:** The number of wheat leaf rust samples and isolates obtained from different growing governorates in Egypt, during the 2019/20 and 2020/21 growing seasons.

No.	Governorate	Season/Number of Samples/Isolates per Governorate				Total	
		2019/20		2020/21			
		No. of Samples	No. of Isolates	No. of Samples	No. of Isolates	No. of Samples	No. of Isolates
1	Kafrelsheikh	25	19	22	15	47	34
2	Beheira	22	17	20	14	42	31
3	Sharkia	27	20	25	17	52	37
4	Alexandria	20	15	18	13	38	28
Total		94	71	85	59	179	130

**Table 2 biology-10-00925-t002:** Disease severity (%) of 16 Egyptian wheat cultivars of leaf rust in four governorates, during 2019/20 and 2020/21 growing seasons.

No.	Cultivar	Governorates/Seasons/Disease Severity							
		Kafrelsheikh		Beheira		Sharkia		Alexandria	
		2019/20	2020/21	2019/20	2020/21	2019/20	2020/21	2019/20	2020/21
1	Gemmeiza-7	60S ^a^	70S	70S	80S	80S	90S	70S	90S
2	Gemmeiza-9	5R ^b^	5MR	10MR	5R	5MS	10S	SS	5S
3	Gemmeiza-10	5MS	5MR	10MS	5S	5S	10S	5R	5MR
4	Gemmeiza-11	70S	80S	60S	90S	80S	80S	70S	90S
5	Gemmeiza-12	5R	5MR	TrR	TrMR	5MR	5R	5R	5MR
6	Sakha-93	50S	80S	40S	70S	70S	70S	60S	80S
7	Sakha-94	5MR ^d^	5MS	5MR	5R	5MS	10MS	TrMS	5MS
8	Sakha-95	5MR	TrMR	10MR	10R	5S	5R	5MS	5S
9	Sids-1	40S	60S	50S	70S	70S	50S	80S	80S
10	Sids-12	5MS ^e^	10MS	5S	5MS	5S	5MS	10MS	5S
11	Sids-13	5MR	10MR	10R	5S	10S	10S	5S	10S
12	Sids-14	5MR	5S	10R	20MS	10MS	10S	5MS	10S
13	Misr-1	10R	5MR	TrMR	5R	5MS	5MS	5MS	5MR
14	Misr-2	5R	TrR ^c^	TrMR	TrR	5MS	5MR	TrMR	TrMR
15	Misr-3	TrMR	TrR	5MR	TrMR	5MR	5R	TrMR	TrR
16	Shandweel-1	5MS	5S	5MS	5S	10MS	5MS	5S	5S

^a^ S = susceptible, ^b^ R = resistant, ^c^ TrR = trace resistant, ^d^ MR = moderately resistant, ^e^ MS = moderately susceptible.

**Table 3 biology-10-00925-t003:** The number of isolates and frequency (%) of Egyptian wheat cultivars in different governorates, during 2019/20 and 2020/21 growing seasons.

Cultivar	Kafrelsheikh				Beheira				Sharkia				Alexandria				Total			
	2019/2020		2020/2021		2019/2020		2020/2021		2019/2020		2020/2021		2019/2020		2020/2021		2019/2020		2020/2021	
	NI ^a^	FR ^b^	NI	FR	NI	FR	NI	FR	NI	FR	NI	FR	NI	FR	NI	FR	NI	FR	NI	FR
Gemmeiza-7	4	21.1	2	13.3	4	23.5	1	7.1	2	10.0	2	12.0	1	6.7	1	7.7	11	15.5	6	10.2
Gemmeiza-9	0	0.0	0	0.0	0	0.0	0	0.0	1	5.0	1	5.9	0	0.0	1	7.7	1	1.4	2	3.4
Gemmeiza-10	2	10.5	0	0.0	1	5.9	2	14.3	1	5.0	1	5.9	0	0.0	0	0.0	4	5.6	3	5.1
Gemmeiza-11	4	21.1	3	20.0	4	23.5	3	21.4	2	10.0	2	12.0	1	6.7	1	7.7	11	15.5	9	15.3
Gemmeiza-12	0	0.0	0	0.0	0	0.0	0	0.0	0	0.0	0	0.0	0	0.0	0	0.0	0	0.0	0	0.0
Sakha-93	2	10.5	3	20.0	3	17.7	2	14.3	2	10.0	2	12.0	2	13.0	1	7.7	9	12.7	8	13.6
Sakha-94	0	0.0	1	6.7	0	0.0	0	0.0	1	5.0	1	5.9	1	6.7	1	7.7	2	2.8	3	5.1
Sakha-95	0	0.0	0	0.0	0	0.0	0	0.0	1	5.0	0	0.0	1	6.7	1	7.7	2	2.8	1	1.7
Sids-1	3	15.8	2	13.3	2	11.8	2	14.3	3	15.0	2	12.0	2	13.0	2	15.0	10	14.1	8	13.6
Sids-12	2	10.5	2	13.3	2	11.8	1	7.1	1	5.0	1	5.9	1	6.7	1	7.7	6	8.5	5	8.5
Sids-13	0	0.0	0	0.0	0	0.0	1	7.1	1	5.0	1	5.9	2	13.0	2	15.0	3	4.2	4	6.8
Sids-14	0	0.0	1	6.7	0	0.0	1	7.1	2	10.0	1	5.9	2	13.0	1	7.7	4	5.6	4	6.8
Misr-1	0	0.0	0	0.0	0	0.0	0	0.0	1	5.0	2	12.0	1	6.7	0	0.0	2	2.8	2	3.4
Misr-2	0	0.0	0	0.0	0	0.0	0	0.0	1	5.0	0	0.0	0	0.0	0	0.0	1	1.4	0	0.0
Misr-3	0	0.0	0	0.0	0	0.0	0	0.0	0	0.0	0	0.0	0	0.0	0	0.0	0	0.0	0	0.0
Shandweel-1	2	10.5	1	6.7	1	5.9	1	7.1	1	5.0	1	12.0	1	6.7	1	7.7	5	7.1	4	6.8
Total	19	-	15	-	17	-	14	-	20	-	17	-	15	-	13	-	71	-	59	-

^a^ NI = number of isolates, ^b^ FR = frequency (%).

**Table 4 biology-10-00925-t004:** Leaf rust races, virulence formula, number of isolates, and frequency (%) of *P. triticina* collected from wheat cultivars cultivated in various wheat-growing areas in Egypt, during 2019/20 and 2020/21. Monitoring of leaf rust virulence pattern through virulence formula was done on the basis of reaction types in a host pathogen system.

No.	Race	Virulence Formula	Season/No. of Isolates and Frequency (%)				Total	
			2019/2020		2020/2021			
			NI ^a^	FR ^b^	NI	FR	NI	FR
1	GTTPT	2a, 9, 16, 24, 26, 3Ka, 11, 17, 30, 10, 21, 2b, 14b, 15,36, 42	6	8.45	3	5.08	9	6.92
2	HTTDS	2a, 3, 9, 16, 24, 26, 3Ka, 11, 17, 30, 21, 14b, 15, 36	3	4.23	4	6.78	7	5.38
3	LGDTT	1, 9, 16, 24, 26, 17, 10, 18, 21, 2b, 14b, 15, 36, 42	2	2.82	1	1.69	3	2.31
4	LTCGT	1, 9, 16, 24, 26, 30, 18, 14b, 15, 36, 42	1	1.41	2	3.39	3	2.31
5	LTTFT	1, 9, 16, 24, 26, 3Ka, 11, 17, 30, 21, 14b, 15, 36, 42	1	1.41	1	1.69	2	1.54
6	MBGJT	1, 3, 11, 18, 21, 14b, 15, 36, 42	1	1.41	-	0.00	1	0.77
7	MTTGT	1,3, 9, 16, 24, 26, 3Ka, 11, 17, 30, 18, 14b, 15, 36, 42	14	19.70	7	11.90	21	16.20
8	NRNJT	1, 2c, 9, 16, 26, 3Ka, 17, 18, 21, 14b, 15, 36, 42	1	1.41	2	3.39	3	2.31
9	PBPPP	1, 2c, 3, 3Ka, 17, 30, 10, 21, 2b, 14b, 36, 42	3	4.23	1	1.69	4	3.08
10	PKKSR	1, 2c, 3, 16, 24, 26, 11, 17, 30, 10, 18, 21, 14b, 15, 42	2	2.82	1	1.69	3	2.31
11	PKTTS	1, 2c, 3, 16, 24, 26, 3Ka, 11, 17, 30, 10, 18, 21, 2b, 14b, 15,36	3	4.23	-	0.00	3	2.31
12	PTKKT	1, 2c, 3, 9, 16, 24, 26, 11, 17, 30, 18, 21, 2b, 14b, 15, 36, 42	4	5.63	2	3.39	6	4.62
13	PTTJT	1, 2c, 3, 9, 16, 24, 26, 3Ka, 11, 17, 30, 18, 21, 14b, 15, 36, 42	2	2.82	4	6.78	6	4.62
14	STSJT	1, 2a, 2c, 9, 16, 24, 26, 3Ka, 11,17, 18, 21, 14b, 15, 36, 42	10	14.10	9	15.30	19	14.60
15	STTLT	1, 2a, 2c, 9, 16, 24, 26, 3Ka, 11, 17, 30, 10, 14b, 15, 36, 42	2	2.82	5	8.47	7	5.38
16	TTTKS	1, 2a, 2c, 3, 9, 16, 24, 26, 3Ka, 11, 17, 30, 18, 21, 2b, 14b, 15, 36	10	14.10	7	11.90	17	13.10
17	TTTST	1, 2a, 2c, 3, 9, 16, 24, 26, 3Ka, 11, 17, 30, 10, 18, 21, 14b, 15, 36, 42	6	8.45	10	16.90	16	12.30
Total			71	100.0	59	100.0	130	100.0

^a^ NI = Number of isolates, ^b^ FR = Frequency (%).

**Table 5 biology-10-00925-t005:** Physiological leaf rust races identified in the Egyptian wheat cultivars, in different governorates, during 2019/20 and 2020/21 growing seasons.

Cultivar	Governorate/Leaf Rust Races			
	Kafrelsheikh	Beheira	Sharkia	Alexandria
2019/2020 Season				
Gemmeiza-7	HTTDS STSJT	MTTGT -	MTTGT -	STSJT -
Gemmeiza-9	-	-	TTTST	-
Gemmeiza-10	LGDTT	MBGJT	TTTST	-
Gemmeiza-11	TTTKS	TTTKS	MTTGT	TTTST
Gemmeiza-12	-	-	-	-
Sakha-93	PBPPP PTTJT	PBPPP PTTJT	PTKKT -	LTTFT PTKKT
Sakha-94	-	-	TTTST	TTTST
Sakha-95	-	-	PTKKT	TTTST
Sids-1	HTTDS STSJT	STSJT LTCGT	MTTGT PKTTS	PKTTS MTTGT
Sids-12	STSJT	STSJT	MTTGT	MTTGT
Sids-13	- -	- -	PKKSR -	NRNJT PKKSR
Sids-14	- -	- -	TTTKS STTLT	TTTKS STTLT
Misr-1	-	-	GTTPT	GTTPT
Misr-2	-	-	GTTPT	-
Misr-3	-	-	-	-
Shandweel-1	GTTPT	GTTPT	MTTGT	MTTGT
2020/2021 season				
Gemmeiza-7	STSJT HTTDS	STSJT -	MTTGT -	HTTDS -
Gemmeiza-9	-	-	TTTST	TTTST
Gemmeiza-10	- -	LGDTT TTTKS	TTTST	-
Gemmeiza-11	PTTJT TTTKS	TTTKS -	TTTST -	TTTST -
Gemmeiza-12	-	-	-	-
Sakha-93	STSJT PTTJT	PTTJT PBPPP	PTKKT PTTJT	LTTFT -
Sakha-94	TTTKS	-	TTTST	TTTST
Sakha-95	-	-	-	TTTST
Sids-1	STSJT LTCGT	STSJT LTCGT	MTTGT HTTDS	HTTDS MTTGT
Sids-12	STSJT	STSJT	MTTGT	MTTGT
Sids-13	- -	STTLT -	NRNJT -	NRNJT PKKSR
Sids-14	STTLT	STTLT	STTLT	STTLT
Misr-1	- -	- -	GTTPT PTKKT	- -
Misr-2	-	-	-	-
Misr-3	-	-	-	-
Shandweel-1	GTTPT	GTTPT	MTTGT	TTTST

**Table 6 biology-10-00925-t006:** Number and frequency (%) of *Puccinia triticina* virulent isolates to 22 *Lr* genes during 2019/2020 and 2020/2021 seasons at the seedling stage.

No.	*Lr* Genes	Season/No. of Isolates and Virulence Frequency (%)				Total	
		2019/2020		2020/2021			
		NI ^a^	FR ^b^	NI	FR	NI	FR
1	*Lr1*	62	87.32	52	88.14	114	87.69
2	*Lr2a*	37	52.11	38	64.41	75	57.69
3	*Lr2b*	28	39.44	14	23.73	42	32.31
4	*Lr2c*	43	60.56	41	69.49	84	64.62
5	*Lr3*	48	67.61	36	61.02	84	64.62
6	*Lr3ka*	59	83.1	49	83.05	108	83.08
7	*Lr9*	62	87.32	57	96.61	119	91.54
8	*Lr10*	27	38.03	26	44.07	53	40.77
9	*Lr11*	64	90.14	53	89.83	117	90.00
10	*Lr14b*	71	100.00	59	100.00	130	100.00
11	*Lr15*	68	95.77	58	98.31	126	96.92
12	*Lr16*	67	94.37	58	98.31	125	96.15
13	*Lr17*	69	97.18	57	96.61	126	96.92
14	*Lr18*	57	80.28	46	77.97	103	79.23
15	*Lr19*	0	0.00	0	0.00	0	0.00
16	*Lr21*	54	76.06	45	76.27	99	76.15
17	*Lr24*	67	94.37	58	98.31	125	96.15
18	*Lr26*	43	60.56	40	67.80	83	63.85
19	*Lr28*	0	0.00	0	0.00	0	0.00
20	*Lr30*	57	80.28	47	79.66	104	80.00
21	*Lr36*	69	97.18	58	98.31	127	97.69
22	*Lr42*	52	73.24	42	71.19	94	72.31
Total		71	-	59	-	130	-

^a^ NI = number of isolates, ^b^ FR = frequency (%).

## Data Availability

Not applicable.

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
