# Peer review of "The Emergence of New Aggressive Leaf Rust Races with the Potential to Supplant the Resistance of Wheat Cultivars"

_biology, 2021, doi:10.3390/biology10090925_

Round 1

Reviewer 1 Report

I have read the manuscript “The emergence of new aggressive leaf rust races with the potential to supplant the resistance of wheat cultivars” by Omara et al. This article analyses the physiological races of leaf rust from Egyptian commercial wheat cultivars. They also studied the geographic distribution of P. triticina populations in four main wheat-growing governorates in Egypt during two successive seasons (2019/2020 and 2020/2021). This is a very detailed study and the findings are extremely important. I think this work could be published in Biology, but there are a few comments and questions, which I mention below, that authors should address in order to accept this manuscript.

P-1L-21, Grown instead of growing

P-1L-26, delete ‘were

P-1L-28, delete ‘to the entire globe’

P-1L-29, delete ‘as well as helping’ and write …..and help

L-75, delete is, and write ‘This might mainly be…………

L-79, write ‘might be considered hotspots…………

L-114-117, change the statement as ‘In the current study, our objectives were to define the physiological races of leaf rust that overcome the resistance in Egyptian commercial wheat cultivars and to understand the pathotype-cultivar-location association and its role in susceptibility/resistance of new wheat cultivars to P. triticina.’

L149-155, Inoculation and purification process is not very clear. Describe more elaborately to get single pustule isolates.

L-167, How many replicates of each NIL were used during inoculation by each isolate? How many days after infection, you assessed the disease? 

L-169, delete ‘normal

You have discussed the effect of location-cultivars association on the number of pathotypes and their frequency. How about temporal or seasonal variation?

You can place Figures 6 and 8 as supplementary files

Gemmeiza-12 and Misr-3 cultivars were resistant in all tested locations in two seasons, with no disease symptoms. What could be probable reasons for the durable high immune reaction? Are these cultivars the representative examples of natural gene pyramiding in wheat?

Finally, could you spend a few words on the role of Lr9 in resistance against rust?

Author Response

Thank you very much for your kindness.

P-1L-21, Grown instead of growing

Done.

P-1L-26, delete ‘were

Done.

P-1L-28, delete ‘to the entire globe’

Done.

P-1L-29, delete ‘as well as helping’ and write …..and help

Done.

L-75, delete is, and write ‘This might mainly be…………

Done.

L-79, write ‘might be considered hotspots…………

Done.

L-114-117, change the statement as ‘In the current study, our objectives were to define the physiological races of leaf rust that overcome the resistance in Egyptian commercial wheat cultivars and to understand the pathotype-cultivar-location association and its role in susceptibility/resistance of new wheat cultivars to P. triticina.’

Done.

L149-155, Inoculation and purification process is not very clear. Describe more elaborately to get single pustule isolates.

This paragraph was improved and simply explained.

L-167, How many replicates of each NIL were used during inoculation by each isolate? How many days after infection, you assessed the disease? 

We collected 3-5 pustules from each sample and the disease was assessed 12 days after inoculation.

L-169, delete ‘normal

Done.

You have discussed the effect of location-cultivars association on the number of pathotypes and their frequency. How about temporal or seasonal variation?

Thank you very much. Our future work will concentrate on temporal variation effects on the number of pathotype.

You can place Figures 6 and 8 as supplementary files

Thank you for the suggestion but we think these figures are important in the manuscript.

Gemmeiza-12 and Misr-3 cultivars were resistant in all tested locations in two seasons, with no disease symptoms. What could be probable reasons for the durable high immune reaction? Are these cultivars the representative examples of natural gene pyramiding in wheat?

It could be due to the existence of different resistance genes and gene pyramiding in these cultivars which improved their resistance to the disease.

Finally, could you spend a few words on the role of Lr9 in resistance against rust?

Done. Please check Line768-Line770.

Reviewer 2 Report

This paper describes the identification of various pathotypes of leaf rust isolated from 4 different wheat cultures across Egypt.  The authors also assessed various cultivars of wheat for leaf rust resistance found from each region.  They conducted principal component and hierarchal analysis to find groups of "races" that clustered together.  Using RAPD markers, the authors isolated DNA and found various bands that were associated with pathotypes. Furthermore the authors uses isogenic lines of wheat with different Lr genes to test for infection levels and infection frequency of different leaf rust races.   LR19 and LR28 were found to be resistant to leaf rust.   Although I don't detect any plagarism with this article, the content and methods are almost identical to that found in Nemati et al. 2019 (Virulence of Leaf Rust Physiological Races in Iran from 2010 to 2017) although this paper is not cited in this manuscript.

Overall, the scientific process is sound and I don't see any major issues with the methodologies.  However, I found this paper to be very confusing and not written clearly.  Many of the tables were busy with too much information.  The methods did not have clear headings and the results were too detailed and did not highlight the main findings of the paper. 

I have a few suggestions for improving this paper which would make it suitable for publication. 

Line 12:  reword to "The pathogen that causes wheat leaf rust"

Line 18:  I struggled with the word "races" for pathogenicity.  Could you please define what is considered a specific race for fungal rusts?  This is for publication in Biology and most biologists would define race as a subspecies with specific genetic differences. 

Line 24:  The races should not be classified in 3 groups (150, 200, 300, etc), but had DNA bands with a 300 Bp marker obtained from RAPD.  Again, this is confusing. 

Line 41:  delete "Still exercises....of the"

Line 64:  delete the severity and high and replace with "then susceptibility to"

Line 69:  Provide the common name for the alternative host of P. triticinia. 

Line 70: italicize all scientific names

Line 86:  Replace "established" with "identified"

Line 90: Define LR genes and give examples of them in other plants.  What are their functions?  Are the R-proteins or other enzymes?  You need 1-2 paragraphs describing these. 

Lines 95-100: Delete this paragraph.  It is redundant.

Line 105: Define RAPD the first time you abbreviate it. 

Line 104-112:  This paragraph is not worded clearly.  Are you describing other studies?  This seems out of place and read more like a result than an intro. 

Line 125:  Can you provide a map of Egypt showing where these governorates are?  For those of us not very familiar with Egypt's geography, it is difficult to discern if these are in different locations, climates (ie. along the river vs. desert). Why were these 4 regions chosen?

Line 139:  What does "Check" mean?

Methods overall:  I think being more descriptive with the headings (2.1.2) Sampling of Wheat Samples across 4 areas of Egypt, etc.  Ex. 2.3.1 Fungal DNA Isolation and Ampliflication using RADP Primers. 

Line 171, 172, 173:  Why is (IT) in parenthesis near all three infection levels?

Results:  Is it possible to streamline the methods and make them shorter?

Figure 1:  The caption D should read "Number of isolates for each region"

Table 2:  You do not need the letters a, b, c, d.  These are confusing.  Also, what does Tr mean?  This is not indicated in the caption.

Table 3:  This looks like figure 1 in a table format.  Do you need both?

Table 344-346:  Reword.  Also, the way you describe giving 5 letter designations for each pathotype/race is very confusion.  Can you rewrite this and describe how the 5-letter naming designation is applied?

Table 4:  You need spaces between your text and your tables/figures. 

Table 4:  Please explain what virulence formula is. 

Figure 7 needs a better explanation for a clearer way of presenting the data. What are the values on the y axis? 

Lines 561:  Please be clear.  Are you using isogenic lines of wheat that specifically have the Leaf Rust Resistance gene 19 etc.?  If so, be clear.

Lines 636-653;  Parts of this paragraph belong in the introduction, not the discussion.  The discussion could use a better explanation of why these results are important and how they compare to other studies (for example, Nemati et al. 2019 which does a similar study). 

Lines 700-708:  This needs to be reworded.  It is very unclear. 

Lines 708-720:  Are you explaining your results or the results of other studies?  It is difficult to discern this with the current text. 

Author Response

Thank you very much for your kindness.

Although I don't detect any plagiarism with this article, the content and methods are almost identical to those found in Nemati et al. 2019 (Virulence of Leaf Rust Physiological Races in Iran from 2010 to 2017) although this paper is not cited in this manuscript.

Thank you very much. Nemati et al. 2019 was added to the text and reference list.

Overall, the scientific process is sound and I don't see any major issues with the methodologies.  However, I found this paper to be very confusing and not written clearly.  Many of the tables were busy with too much information.  The methods did not have clear headings and the results were too detailed and did not highlight the main findings of the paper. 

I have a few suggestions for improving this paper which would make it suitable for publication. 

Line 12:  reword to "The pathogen that causes wheat leaf rust"

Done.

Line 18:  I struggled with the word "races" for pathogenicity.  Could you please define what is considered a specific race for fungal rusts?  This is for publication in Biology and most biologists would define race as a subspecies with specific genetic differences. 

We added the definition of race in Line 68.

Line 24:  The races should not be classified in 3 groups (150, 200, 300, etc), but had DNA bands with a 300 Bp marker obtained from RAPD.  Again, this is confusing. 

We deleted the confusing part from the sentence.

Line 41:  delete "Still exercises....of the"

Done.

Line 64:  delete the severity and high and replace with "then susceptibility to"

Done.

Line 69:  Provide the common name for the alternative host of P. triticinia

We added the Yellow Meadow-rue to the text.

Line 70: italicize all scientific names

Done.

Line 86:  Replace "established" with "identified"

Done.

Line 90: Define LR genes and give examples of them in other plants.  What are their functions?  Are the R-proteins or other enzymes?  You need 1-2 paragraphs describing these. 

Done.

Lines 95-100: Delete this paragraph.  It is redundant.

Done.

Line 105: Define RAPD the first time you abbreviate it. 

We added random amplified polymorphic DNA.

Line 104-112:  This paragraph is not worded clearly.  Are you describing other studies?  This seems out of place and read more like a result than an intro. 

We deleted some sentences from this paragraph and kept only the most important parts that show the reason for using RAPD analysis by utilizing the results of previous studies.

Line 125:  Can you provide a map of Egypt showing where these governorates are?  For those of us not very familiar with Egypt's geography, it is difficult to discern if these are in different locations, climates (ie. along the river vs. desert). Why were these 4 regions chosen?

Done, please check Figure 9.

Line 139:  What does "Check" mean?

It means the highly susceptible cultivars. We deleted the word check.

Methods overall:  I think being more descriptive with the headings (2.1.2) Sampling of Wheat Samples across 4 areas of Egypt, etc.  Ex. 2.3.1 Fungal DNA Isolation and Amplification using RADP Primers. 

According to the Journal guidelines, the methods should be more descriptive.

Line 171, 172, 173:  Why is (IT) in parenthesis near all three infection levels?

We deleted (IT). No need for it.

Results:  Is it possible to streamline the methods and make them shorter?

Some parts of the methods we rephrased but we kept the details of the methods according to Journal guidelines.

Figure 1:  The caption D should read "Number of isolates for each region"

Done.

Table 2:  You do not need the letters a, b, c, d.  These are confusing.  Also, what does Tr mean?  This is not indicated in the caption.

We deleted a, b, c, d and their meanings. Tr means trace.

Table 3:  This looks like figure 1 in a table format.  Do you need both?

Yes, we need both.

Table 344-346:  Reword.  Also, the way you describe giving 5 letter designations for each pathotype/race is very confusion.  Can you rewrite this and describe how the 5-letter naming designation is applied?

Please check the paragraph of race identification Line 174-Line 182 and Supplementary Table 1.

Table 4:  You need spaces between your text and your tables/figures. 

Done.

Table 4:  Please explain what virulence formula is. 

The virulence formula is the number of broken genes. We added it to table 4.

Figure 7 needs a better explanation for a clearer way of presenting the data. What are the values on the y axis? 

Figure 7 is just summarizing the results of Figure 6 in a simpler way for the reader. It is a dendrogram and so no value.

Lines 561:  Please be clear.  Are you using isogenic lines of wheat that specifically have the Leaf Rust Resistance gene 19 etc.?  If so, be clear.

Done. Modified.

Lines 636-653;  Parts of this paragraph belong in the introduction, not the discussion.  The discussion could use a better explanation of why these results are important and how they compare to other studies (for example, Nemati et al. 2019 which does a similar study). 

Nemati et al., was added to references and this paragraph explain the reason for conducting the survey and then we explained the results of our manuscript.

Lines 700-708:  This needs to be reworded.  It is very unclear. 

We wrote the paragraph again in a clearer way.

Lines 708-720:  Are you explaining your results or the results of other studies?  It is difficult to discern this with the current text. 

We discussed why we used RAPD and reported our results and compared with other studies.

Round 2

Reviewer 2 Report

The authors addressed many of my questions, however, I still think there are parts of the manuscript that are confusing and could use simple rewording to make it clearer to the reader. 

For example.

Line 26:  The races were classified into 3 groups, based on their distinct DNA band sizes (either 150bp, 200bp, and 300bp) after RAPD analysis.

Line 78:  a better for for important would be "virulent"

Line 99:  a better for for consequent would be "consistent"

Line 105:  This is one of the areas that I mentioned was confusing in my first review.  Although the authors rewrote some of it, it is still not clear.  A suggestion would be to put at the beginning of the paragraph starting line 105... "According to a study done by Kolmer et al, (2000) (27), t...he virulence/avirulence....     By stating it this way, the reader is not confused whether you are describing this current study or one that was done by another researcher. 

Line 118: Delete "We believe that". 

Table 2:  Please indicate in a caption or heading what R, MS, Tr, etc. mean. 

You still need spaces between some of your paragraphs and new headings.  

Table 4:  In my initial review I asked to define or describe what virulence formula was.  You noted that it was broken genes.  However, I still don't see this in the manuscript and still don't have a great understanding of virulence formula based on your description. 

Lines 696 (whole paragraph).  I have the same concern with this section that I did with the introduction.  From your description it is difficult to ascertain whether you did this study or it was another research.  One way to make this clear would be to use wording....  for example "Previous research has shown that the differences between the races of P...  You say this in the next sentence, but I would start the paragraph off with it. 

Author Response

Comments and Suggestions for Authors

The authors addressed many of my questions, however, I still think there are parts of the manuscript that are confusing and could use simple rewording to make it clearer to the reader. 

Thank you very much for your kindness and valuable suggestions.

Line 26:  The races were classified into 3 groups, based on their distinct DNA band sizes (either 150bp, 200bp, and 300bp) after RAPD analysis.

DONE.

Line 78:  a better for important would be "virulent"

DONE.

Line 99:  a better for consequent would be "consistent"

DONE.

Line 105:  This is one of the areas that I mentioned was confusing in my first review.  Although the authors rewrote some of it, it is still not clear.  A suggestion would be to put at the beginning of the paragraph starting line 105... "According to a study done by Kolmer et al, (2000) (27), t...he virulence/avirulence....     By stating it this way, the reader is not confused whether you are describing this current study or one that was done by another researcher. 

DONE.

Line 118: Delete "We believe that". 

DONE.

Table 2:  Please indicate in a caption or heading what R, MS, Tr, etc. mean.

DONE.

You still need spaces between some of your paragraphs and new headings.  

DONE.

Table 4:  In my initial review I asked to define or describe what virulence formula was.  You noted that it was broken genes.  However, I still don't see this in the manuscript and still don't have a great understanding of virulence formula based on your description. 

DONE. We added it to the title of the Table. Please also find in the methods Line 171.

Lines 696 (whole paragraph).  I have the same concern with this section that I did with the introduction.  From your description it is difficult to ascertain whether you did this study or it was another research.  One way to make this clear would be to use wording....  for example "Previous research has shown that the differences between the races of P...  You say this in the next sentence, but I would start the paragraph off with it. 

DONE. We put in the current study before our work and we put previous studies showed that before citing the others work.
